# Characterizing the dynamics, reactivity and controllability of moods in depression with a Kalman filter

**Jolanda Malamud**[1]*, **Sinan Guloksuz**[2,3], **Ruud van Winkel**[2,4], **Philippe Delespaul**[2], **Marc A. F. De Hert**[4,5,6,7], **Catherine Derom**[8,9], **Evert Thiery**[10], **Nele Jacobs**[2,11], **Bart P. F. Rutten**[2], **Quentin J. M. Huys**[1]

**1** Applied Computational Psychiatry Lab, Mental Health Neuroscience Department, Division of Psychiatry and Max Planck Centre for Computational Psychiatry and Ageing Research, Queen Square Institute of Neurology, University College London, London, United Kingdom, **2** Department of Psychiatry and Neuropsychology, School for Mental Health and Neuroscience, Maastricht University Medical Centre, Maastricht, The Netherlands, **3** Department of Psychiatry, Yale School of Medicine, New Haven, Connecticut, United States of America, **4** Department of Neurosciences, Centre for Clinical Psychiatry, KU Leuven, Leuven, Belgium, **5** Department of Psychotic Disorders, University Psychiatric Centre KU Leuven, Kortenberg, Belgium, **6** Leuven Brain Institute, KU Leuven, Leuven, Belgium, **7** Antwerp Health Law and Ethics Chair, University of Antwerp, Antwerp, Belgium, **8** Centre of Human Genetics, University Hospitals Leuven, KU Leuven, Leuven, Belgium, **9** Department of Obstetrics and Gynecology, Ghent University Hospitals, Ghent University, Ghent, Belgium, **10** Department of Neurology, Ghent University Hospital, Ghent University, Ghent, Belgium, **11** Faculty of Psychology, Open University of the Netherlands, Heerlen, The Netherlands

* j.malamud@ucl.ac.uk

**Data Availability Statement:** The code for data analysis of this study is available from a Github repository: github.com/huyslab/esmssm_public.

## Abstract

### Background

Mood disorders involve a complex interplay between multifaceted internal emotional states, and complex external inputs. Dynamical systems theory suggests that this interplay between aspects of moods and environmental stimuli may hence determine key psycho-pathological features of mood disorders, including the stability of mood states, the response to external inputs, how controllable mood states are, and what interventions are most likely to be effective. However, a comprehensive computational approach to all these aspects has not yet been undertaken.

### Methods

Here, we argue that the combination of ecological momentary assessments (EMA) with a well-established dynamical systems framework—the humble Kalman filter—enables a comprehensive account of all these aspects. We first introduce the key features of the Kalman filter and optimal control theory and their relationship to aspects of psychopathology. We then examine the psychometric and inferential properties of combining EMA data with Kalman filtering across realistic scenarios. Finally, we apply the Kalman filter to a series of EMA datasets comprising over 700 participants with and without symptoms of depression.

**Funding:** JM was supported by an International Max Planck Research School on Computational Methods in Psychiatry and Ageing Research (IMPRS COMP2PSYCH) and a Wellcome Trust grant to QJMH (221826/Z/20/Z). In addition, we acknowledge support by the UCLH NIHR BRC to JM. The Twinscan project was part of the the European Network of National Schizophrenia Networks Studying Gene-Environment Interactions (EU-GEI) Project, which was funded by grant agreement HEALTH-F2-2010-241909 (Project EU-GEI) from the European Community's Seventh Framework Programme. The funders had no role in study design, data collection and analysis, decision to publish, or preparation of the manuscript.

**Competing interests:** I have read the journal's policy and the authors of this manuscript have the following competing interests: QJMH has obtained a research grant from Koa Health, and obtained fees and options for consultancies for Aya Technologies and Alto Neuroscience. All other authors report no conflicts of interest.

## Results

The results show a naive Kalman filter approach performs favourably compared to the standard vector autoregressive approach frequently employed, capturing key aspects of the data better. Furthermore, it suggests that the depressed state involves alterations to interactions between moods; alterations to how moods responds to external inputs; and as a result an alteration in how controllable mood states are. We replicate these findings qualitatively across datasets and explore an extension to optimal control theory to guide therapeutic interventions.

## Conclusions

Mood dynamics are richly and profoundly altered in depressed states. The humble Kalman filter is a well-established, rich framework to characterise mood dynamics. Its application to EMA data is valid; straightforward; and likely to result in substantial novel insights both into mechanisms and treatments.

## Author summary

In this study, we aimed to understand the dynamics of mood in the context of depression, utilizing experience sampling data and well-established mathematical techniques. Our approach sought to overcome limitations of traditional methods and accurately capture the dynamics of moods in real-life situations. Through the application of a Kalman filter to examine mood trajectories in experience sampling data from various datasets, including both patients with depression and healthy controls, we were able to capture the evolution of mood, interaction among different mood items, and responsiveness to environmental inputs. The study revealed distinct dynamical features characteristic of depression, highlighted the potential of using external factors to influence mood and potentially shift between stable emotional states. The findings offer valuable insights into the impact of depression on mood dynamics and potential intervention strategies, contributing to a better understanding of depression mechanisms. The study also acknowledges the challenges of employing complex models to depict sparse and noisy data, emphasizing the need for further research to address these complexities.

## 1 Introduction

Depression is a common mental disorder that imposes a significant burden on societies worldwide [1]. Its prominence and burden arise from its high prevalence but also from its chronic, recurrent nature [2, 3]. Consequently, a better understanding of the mechanisms underlying the maintenance of the depressed state, and how these are altered by specific interventions, is of critical importance.

Traditionally, conceptualizations of depression—and psychiatric disorders more broadly [4, 5]—have followed the traditional medical model, building on the notion that symptoms are different and independent consequences of an underlying latent disease process [6]. More recently, 'network' conceptualizations have emphasized the complex interaction between the different components (i.a. [6–12]). The attraction of such network models lies in their ability

to capture aspects of the statistical and temporal characteristics of symptoms and other observable features of depression [8, 13–16]. Indeed, network-based statistics, appear to relate richly to aspects of psychopathology, disease progression, and treatment response [17–26]. A robust understanding of how different symptoms or emotions in an individual interrelate may also be therapeutically useful in guiding for instance psychotherapeutic interventions or focus [27].

Network statistics have been derived both from cross-sectional and longitudinal studies which collect data over a period of time and focus more on the individual [28, 29]. Densely sampled longitudinal datasets are becoming increasingly frequent as smartphone-based momentary assessments are becoming more readily available [30–32]. In experience sampling, individuals are asked repeatedly about their current experiences at random times throughout the day [33]. This method stands in contrast to standard psychological questionnaires typically asking individuals to average their symptoms over periods of time which can be unreliable due to recall bias [34]. The in-the-moment assessments reduce such biases, and the repeated sampling increases sensitivity to change over time [35].

There are two main approaches to analyse experience sampling data and related high-density multidimensional longitudinal data: the calculation of summary statistics and a model-based approach. The summary statistics approach involves measures such as instability (root-mean-square successive differences [36]), inertia (temporal autocorrelation [37]), and simpler measures like mean, variance, and cross-correlation [38]. Various studies investigating affect dynamics have demonstrated their importance by showing that altered affect dynamics relate to psychopathology, particularly negative valence [39–42]. Although these statistics are simple and easy to understand, they do not provide a broader framework for understanding and studying mood as they do not capture key aspects of the underlying data generation process.

Model-based analyses typically involve linear regression, whereby experience sampling measures at one time-point are regressed onto previous time-points. Vector autoregressive (VAR) models are the most commonly used, combining multiple summary statistics of mood dynamics into a dynamic framework [19, 43–45]. Nevertheless, VAR approaches have some limitations. First, results from VAR analyses vary considerably depending on methodological choices [15, 46, 47]. Second, VAR models assume that observations are acquired at discrete time points even though a key aspect of experience sampling is the variation of intervals between consecutive samples [48]. Third, VAR models do not allow for measurement errors even though psychological variables are known to be susceptible to measurement error. Unlike dynamic errors which VAR models do allow for, measurement or residual errors occur at only one time-point, but have to be accounted for by the model through persistent noise. Finally, and maybe most critically, prior research has primarily concentrated on affective self-reports in isolation and frequently disregarded the influence of the environmental context [49]. The omission of external stimuli is a critical limitation, as the dynamical properties of a system cannot be fully discerned without knowledge of its inputs. Inputs can profoundly alter the apparent dynamical system [50]. Failing to consider the immediate context in response to which emotions fluctuate may therefore result in incorrect conclusions regarding the underlying affective system. In fact, it is widely recognized that external stimuli have a profound impact on the (observed) time-course of emotional states [50–53].

State space models offer a robust, well-established framework that can handle unequal sampling intervals, missing data, and can account for measurement error and inputs to identify rich dynamical phenomena [9, 54, 55]. These models have been applied across various fields, including ecology [56] and neuroscience [57, 58], and can be functionally equivalent to structural equation models, which are widely utilized in psychology [45, 59–61]. However, despite the increasing theoretical popularity of the dynamic systems perspective [9, 62, 63], state-space

models have not been frequently employed to examine mood time series, especially when incorporating external inputs.

The aim of this paper is to examine a standard, well-characterized state-space model, namely the Kalman filter, as a state space framework for the study of individual differences in mood dynamics. After a brief explanation of state space modeling, we outline how they can be used to address unequal sampling intervals, incorporate external inputs, as well as the techniques we employ to characterize the dynamical system, such as stability and controllability. We then examine theoretical properties of typical experience sampling data acquisition settings. Next, we apply Kalman filtering to three different experience sampling datasets to address several novel questions. First, we verify that Kalman filtering can capture empirical data features such as the mean, the variance within and between mood items, the temporal autocorrelation, and aspects of stability. Second, we examine whether mood dynamics exhibit characteristics specific to depression by analyzing differences between patients with depression and healthy controls, and by studying the relationship between dynamical features and depression symptoms in a large general population sample. Finally, we examine one extended single person dataset during which one patient discontinued medication and experienced worsening symptoms, and use this to examine the potential for Kalman filtering to guide just-in-time treatment interventions. The discussion evaluates the strengths and limitations of the model and offers suggestions for future research.

## 2 Materials and methods

This work was approved by the University College London Research Ethics Committee (REC No 16429/002).

### 2.1 Experience sampling data

Three previously published experience sampling datasets were used.

Dataset 1 [37] comprised experience sampling data, including four mood items (cheerful, content, anxious, and sad), collected over five or six consecutive days. It encompassed two groups: 1) a depressed patient group (N = 150) and 2) a healthy control group (N = 579). Participants received digital wristwatches that prompted them to complete paper-based experience sampling forms 10 times per day. This resulted in a maximum of 50 or 60 measurements of four different mood scores on seven-point Likert scales. After applying exclusion criteria, the final sample included 117 participants in the depressed group and 210 in the healthy group.

Dataset 2 is derived from the TwinssCan project [64] and included 839 participants. The experience sampling measures were administered through a phone application, prompting participants to complete questionnaires about their mood, activities, location, and company. After excluding participants (N = 400) based on the same exclusion criteria as in Dataset 1, the final sample consisted of 439 participants.

Dataset 3 [65] consisted of momentary observations of daily life experiences from a single participant diagnosed with major depression disorder. The participant reported his experiential states 10 times a day over 239 days, including the gradual discontinuation of antidepressant medication. This dataset included three phases: baseline (6 weeks before tapering of antidepressants), discontinuation (8 weeks during discontinuation), and post-transition (6 weeks after the transition occurred).

A detailed description of each dataset is provided in S1 Text, Sec 2 Datasets.

## 2.2 Dynamical modelling

**2.2.1 Kalman filter.** In experience sampling experiments, [37, 64, 65] participants report on $i = 1 \cdots N$ different aspects of their mood at each sampling time point $t$. Typically, around 10 timepoints per day are sampled. To ensure the samples measure the person's state at the time of sampling, the timing of the samples is unpredictable, meaning that time intervals between samples are unequal. Participants may also introduce errors when reporting their mood using the scales. A suitable, simple and tractable model of the temporal structure in such data is a linear Gaussian state space model such as the time-invariant Kalman filter (55, 66; cf. Fig 1A), which postulate an underlying unobserved latent process (the true mood) that evolves as follows:

$$\mathbf{z}_t = \mathbf{A}\mathbf{z}_{t-1} + \mathbf{h} + \mathbf{C}\mathbf{u}_t + \boldsymbol{\epsilon}_t \qquad \boldsymbol{\epsilon}_t \sim \mathcal{N}(0, \boldsymbol{\Sigma}) \tag{1}$$

where $\mathbf{z}_t$ is the true (not directly observable) mood state at time $t$, with the $i$'th component of the vector $[\mathbf{z}_t]_i$ relating to mood $i$. The inclusion of the latent space provides a principled approach to dealing with missing observations, unequal time intervals, and noisy observations. We note it gives rise to two separate noise processes, namely dynamics noise and observation noise. $\mathbf{A}$ is the dynamics matrix of the latent process. Its off-diagonal terms determine how one mood influences other moods. Its diagonal elements determine how a mood persists. The term $\mathbf{h}$ is a bias, i.e. a driving term that is static and determines the 'resting' mood the system

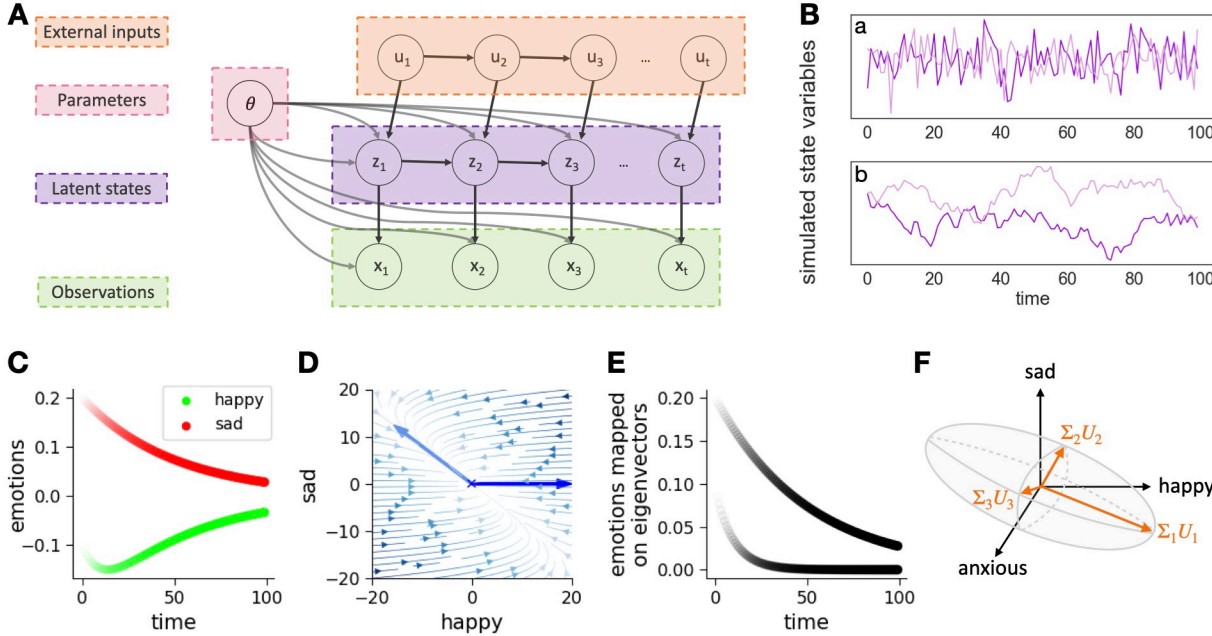

**Fig 1. Dynamical modelling. A)** Graph visualization of the linear dynamical model including external inputs ($\mathbf{u}_t$). $\mathbf{z}_t$ are the latent dynamical states evolving as a Markov process and generating the mood measurements $\mathbf{x}_t$. **B)** Simulated state variables generated by different dynamic matrices: a) shows a fast process generated by a dynamics matrix with eigenvalues close to 0 and b) a slow process generated by a dynamics matrix with eigenvalues close to 1. **C)** shows the trajectories of a two-dimensional system (happy and sad) starting from a randomly chosen initial point. Whereas sad decays independently of happy, the trajectory of happy is more complex and does not straightforwardly decay to zero because it is influenced by the sad variable. However, both variables do converge to zero at some point. **D)** Streamplot demonstrating the evolution of both variables from different starting points. The light blue arrows represent the first eigenvector, while the dark blue arrows represent the second eigenvector. **E)** displays the evolution of the eigenmodes, the independently evolving trajectories resulting from the projecting of the state variables onto the eigenvectors of the dynamics. **F)** The left singular vectors ($U$) of the controllability matrix define an energy ellipsoid where the singular directions corresponding to higher singular values ($\Sigma$) are more controllable. The same input strength $\mathbf{u}$ has a greater impact along the most controllable ($U_1$) than the least controllable direction ($U_3$).

returns to in the absence of any input and noise. The term $\mathbf{u}_t$ denotes the external inputs at time $t$. The inputs are weighted by a matrix $\mathbf{C}$, which determines how strongly each of the input dimensions affects each of the mood items. The external inputs capture the individual's environment, such as their current activities or social interactions. Since these inputs were measured concurrently with the mood items $\mathbf{x}_t$, we made the assumption that the particular input was present and active only at that specific time-point. As environmental stimuli usually have some duration rather than being punctuate, this is an approximation. The Gaussian noise term $\epsilon$ allows the latent state to randomly drift. The Kalman filter assumes that the true internal mood states are observed noisily as follows:

$$\mathbf{x}_t = \mathbf{B}\mathbf{z}_t + \boldsymbol{\eta}_t \qquad \boldsymbol{\eta}_t \sim \mathcal{N}(0, \boldsymbol{\Gamma}) \tag{2}$$

In principle, the Kalman filter allows for observations to represent arbitrary linear mixtures of the latent variables. However, for the present purposes, the matrix $\mathbf{B}$ is set to the identity matrix. This means that each true latent mood item $i$ is noisily observed and that the dynamics matrix $\mathbf{A}$ can be interpreted as interactions between the sampled mood items. We fix $\boldsymbol{\Sigma}$ and $\boldsymbol{\Gamma}$ to be diagonal matrices, meaning that noise in ratings is assumed to be uncorrelated. For simplicity, we also assume Gaussian noise on the observations.

**2.2.2 Unequal sampling intervals.** A key aspect of the experience sampling method is the presence of unequal sampling intervals. The unequal time intervals have mostly been neglected in the analyses of experience sampling data (i.a. [18, 37, 42]), resulting in biases in parameter estimates leading to errors in strength and time course estimates of effects [48, 67]. To take sampling intervals into account, we simply defined fine-grained time steps $\Delta t$ in minutes in the latent state space:

$$\mathbf{x}_{t+\Delta t} = \mathbf{z}_{t+\Delta t} + \sqrt{\boldsymbol{\Gamma}}\boldsymbol{\eta}_{t+\Delta t} \tag{3a}$$

$$\mathbf{z}_{t+\Delta t} = \mathbf{A}^{\Delta t}\mathbf{z}_t + \mathbf{C}\mathbf{u}_{t+\Delta t} + \sum_{k=0}^{\Delta t-1} \mathbf{A}^k[\mathbf{h} + \sqrt{\boldsymbol{\Sigma}}\boldsymbol{\epsilon}_t] \tag{3b}$$

$$\mathbf{A}^* = \mathbf{A}^{\frac{1}{\Delta t}} = \mathbf{V}\mathbf{D}^{\frac{1}{\Delta t}}\mathbf{V}^{\dagger} \tag{3c}$$

This naturally led to sparse observations, which required small dynamics noise and dynamics matrix eigenvalues close to unity (Fig 1B) to avoid overfitting with high-frequency or purely noisy components. The time gap $\Delta t$ between samples is in the exponent of the dynamics matrix $\mathbf{A}$ in Eq 3b. We therefore defined the dynamics matrix as $\mathbf{A}^*$ (cf. 3c) where $\mathbf{V}$ is the matrix of eigenvectors and $\mathbf{D}$ is the diagonal matrix of eigenvalues. This keeps the self-connection diagonal elements of $\mathbf{A}$ close to 1, which ensures slow decay over time. Naturally, this also constrains the dynamics matrix to be dominated by the diagonal, which is expected at small time steps.

**2.2.3 Stability.** A second aspect of interest is the within-person stability of mood, i.e. the intrinsic stability of a person's mood system. In a linear state space model without external manipulations and noise, the system converges to a fixed point around the average or mean $\mathbf{z}^{\text{FP}} = (\mathbf{I} - \mathbf{A})^{-1}\mathbf{h}$ (which is at the origin in the absence of a bias $\mathbf{h}$).

The momentary changes are described by the dynamics matrix $\mathbf{A}$. The eigenmodes of the system $\tilde{\mathbf{z}}$ are identified by projecting $\mathbf{z}$ onto the eigenvectors of $\mathbf{A}$. These eigenmodes evolve independently without interaction and represent the underlying emotional 'factor' combinations that drive an individual's affective state changes over time. The eigenvector with the highest eigenvalue denotes the most stable emotional combination, i.e. the one which changes most slowly. The eigenvector with the smallest eigenvalue denotes the most transitory and

least persistent mood mode (cf. Fig 1C–1E). The overall tendency to move quickly or slowly (the overall stability) was examined by computing the determinant $|\mathbf{A}|$. Our focus here was on the real parts of the eigenspectra.

We note here the importance of correctly considering inputs $\mathbf{u}$. If there is an input, this fixed point moves to $\mathbf{z}_t^{\mathrm{FP}} = (\mathbf{I} - \mathbf{A})^{-1}(\mathbf{h} + \mathbf{C}\mathbf{u}_t)$, which can be arbitrarily far away from the fixed point without input. Similarly, if there are inputs which are not considered when inferring $\mathbf{A}$, then changes due to inputs will be ascribed to the internal dynamics.

**2.2.4 Controllability.** Mood depends on external inputs, both in and out of the lab [51, 53, 68]. The inclusion of inputs is not only important for correct identification of the intrinsic dynamical aspects of the mood system, but also has interesting potential applications centered around the notion of controllability. The controllability Gramian [69]

$$\mathcal{C} = [\mathbf{C} \ \mathbf{A}\mathbf{C} \ \mathbf{A}^2\mathbf{C} \ \ldots \ \mathbf{A}^{n-1}\mathbf{C}] \tag{4}$$

is an aspect of the system which depends on the two matrices $\mathbf{A}$ and $\mathbf{C}$ and formally describes how 'controllable' a dynamical system is. The definition of control here is the required intensity of the input $\mathbf{u}$ to move the system—the larger the input needs to be, the less controllable the system. Importantly, the Gramian matrix $\mathcal{C}$ has a structure which identifies which combination of emotions can be controlled by which kinds of inputs. The left singular vectors of $\mathcal{C}$ define an energy ellipsoid (Fig 1F). The singular vectors with larger singular values denote more controllable directions. The more controllable a direction is, the less input energy is required to steer the system in that specific direction. An input of a given strength $|\mathbf{u}|$ can move the system further if it aligns with a more controllable direction than a less controllable one.

The notion of control may help identify effective ways to alter an individual's emotional state using external inputs. Indeed, identifying the control inputs $\mathbf{u}$ required to achieve a particular state is the objective of optimal control theory [70]. We explored the use of control theory to identify a sequence of inputs $\mathbf{u}$ to promote positive affective experiences whilst mitigating negative emotional states (see S1 Text, Sec 1.3 Optimal Control for details). Inputs were encoded as one-hot vectors identifying the type of activity, social company and location at the time of affective reports.

**2.2.5 Inference.** Our aim was to recover the parameters $\boldsymbol{\Theta} = \{\mathbf{A}, \mathbf{h}, \mathbf{C}, \boldsymbol{\Sigma}, \boldsymbol{\Gamma}\}$ and to infer the posterior distribution $p(\mathbf{z} \mid \mathbf{x}, \boldsymbol{\Theta})$ over the latent states from individual participants' time series data (here $\mathbf{x}$). This is a standard problem. It involves writing the joint probability of latent states and observations:

$$\log p(\mathbf{z}, \mathbf{x}) = p(\{\mathbf{z}_1, \ldots, \mathbf{z}_T\}, \{\mathbf{x}_1, \ldots, \mathbf{x}_\tau\}) = p(\mathbf{z}_1) \prod_{t=2}^{T} p(\mathbf{z}_t \mid \mathbf{z}_{t-1}) \prod_{t=1}^{\tau} p(\mathbf{x}_t \mid \mathbf{z}_t) \tag{5}$$

where the product is due to the Markov property and other statistical assumptions described above. Taking the log of this leads to a simply computable matrix quadratic form (ignoring the notation and sum over dimensions for clarity):

$$
\begin{aligned}
\log p(\mathbf{z}, \mathbf{x}) \ \ = \ &-\frac{1}{2}\log(2\pi)[\log(|\boldsymbol{\Sigma}|) + (\mathbf{z}_1 - \boldsymbol{\mu}_0 - \mathbf{C}\mathbf{u}_1)^T \boldsymbol{\Sigma}^{-1}(\mathbf{z}_1 - \boldsymbol{\mu}_0 - \mathbf{C}\mathbf{u}_1)] \\
&+ \sum_{t=2}^{T}[\log(|\boldsymbol{\Sigma}|) + (\mathbf{z}_t - \mathbf{A}\mathbf{z}_{t-1} - \mathbf{h} - \mathbf{C}\mathbf{u}_t)^T \boldsymbol{\Sigma}^{-1}(\mathbf{z}_t - \mathbf{A}\mathbf{z}_{t-1} - \mathbf{h} - \mathbf{C}\mathbf{u}_t)] \\
&+ \sum_{t=1}^{\tau}[\log(|\boldsymbol{\Gamma}|) + (\mathbf{x}_t - \mathbf{z}_t)^T \boldsymbol{\Gamma}^{-1}(\mathbf{x}_t - \mathbf{z}_t)]
\end{aligned}
\tag{6}
$$

Using Expectation Maximization [71] we iterated over 1) inferring the latent states using Kalman filtering and smoothing [55, 72], and 2) estimating the parameters by maximizing the joint log-likelihood. These two steps were performed alternately until the marginal likelihood $P(\mathbf{x} \mid \boldsymbol{\Theta})$ converged. For more details see S1 Text, Sec 1.1 Kalman Filter Derivations.

**2.2.6 Identifiability in typical experience sampling scenarios.** The trajectories of latent states can be robustly inferred in these settings [55]. However, for clinical considerations, we were specifically interested in the dynamical parameters of those latent trajectories and therefore required interpretable and recoverable parameters. We examined parameter identifiability in typical experience sampling scenarios with varying data dimensionality (number of emotion self-reports), number of samples, level of noise, and varying temporal sparsity and regularity. We simulated time-series data matching experience sampling statistics and features from Kalman filters with known parameters to study parameter identifiability. Observations were

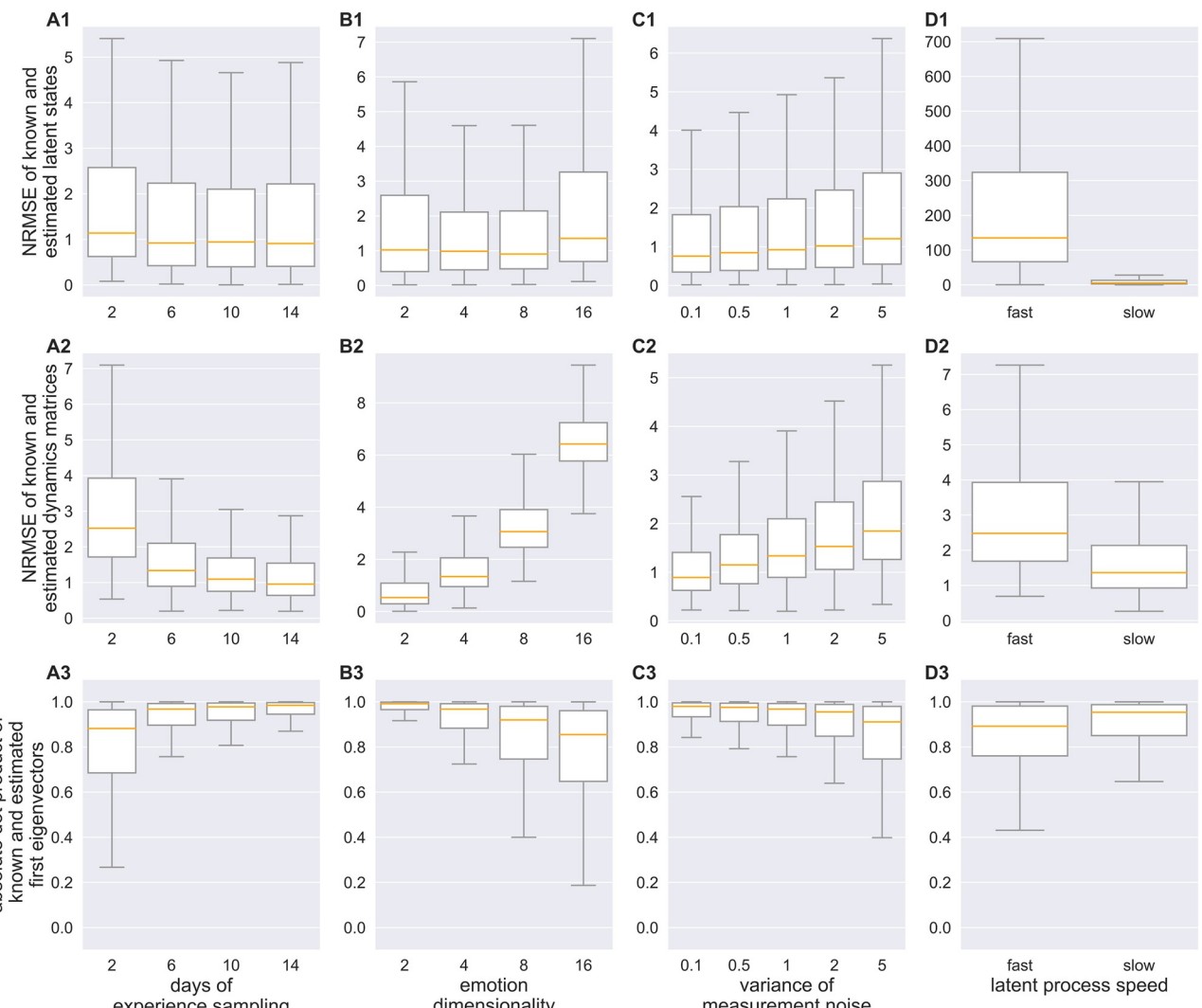

**Fig 2. Identifiability of dynamics in experience sampling setting. A1-D1)** Normalized Root Mean Squared Error (NRMSE) between the known and re-estimated latent states, averaged across emotion trajectories, for different time-series length (A1), emotion dimensionality (B1), measurement noise (C1), and process speed (D1). **A2-D2)** NRMSE between the known and re-estimated dynamics matrix, for varying time-series length (A2), emotion dimensionality (B2), measurement noise (C2), and process speed (D2). **A3-D3)** Absolute dot product between eigenvectors of the known and re-estimated dynamics matrix under the influence of different factors: time-series length (A3), emotion dimensionality (B3), measurement noise (C3), and process speed (D3).

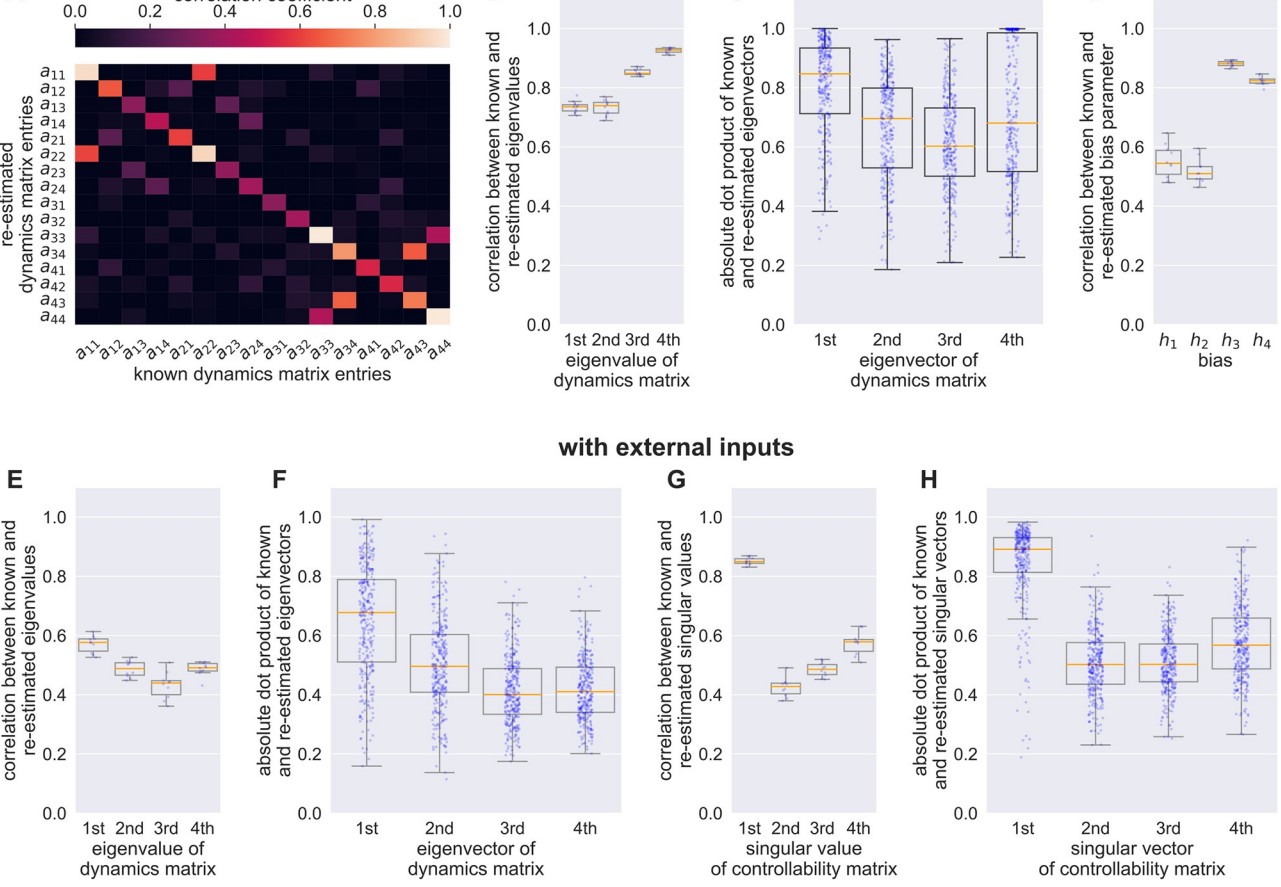

**Fig 3. Parameter recovery. A)** Correlation matrix between true and re-estimated parameters of all single elements of the matrix **A** for the dataset without inputs. The off-diagonal correlations are mostly low. **B)** Correlation between true and re-estimated eigenvalues of the matrix **A**. **C)** Absolute dot product between true and re-estimated eigenvectors of the matrix **A**. Despite the relatively low fidelity in the recovery of individual entries of the matrix **A**, the overall dynamical structure was recovered reasonably in the dataset without inputs. **D)** Correlation between true and re-estimated bias parameters. Different components of the bias parameter were differently well recovered in the data without inputs. **E)** Recovery of eigenvalues of the dynamics matrix **A** in data with inputs. **F)** Recovery of eigenvectors of the dynamics matrix **A** in data with inputs. **G)** Recovery of singular values of the controllability Gramian $\mathcal{C}$ in data with inputs. **H)** Recovery of left singular vectors of the controllability Gramian $\mathcal{C}$ in data with inputs. The dominant component of the controllability Gramian is recovered well.

sampled based on a mixed sampling scheme [73] with semi-random and fixed elements. We defined the number of days and, similar to most experience sampling procedures, sampled 10 beeps per day during selected waking hours, i.e. 9am—11pm. Each beep is randomly scheduled within one of ten blocks throughout the day with at least 30 minutes between consecutive beeps. We simulated 1000 datasets for different numbers of days (days = [2, 6, 10, 14]), different observation noise variance ($\Gamma$ = [0.1, 0.5, 1, 2, 5]) and different numbers of mood items (dimension = [2, 4, 8, 16]). Then we estimated the latent states and parameters for each simulated datasets and calculated similarity measures (Normalized Root Mean Squared Error (NRMSE) and dot product) between the simulated and estimated latent states, as well as between the known parameters used to simulate the datasets and the parameters estimated from the simulated data.

We also examined the recoverability of the parameters in the parameter range corresponding to the datasets employed. This involved estimating parameters from each of the datasets,

generating simulated data using these estimated parameters as true underlying parameters, and then examining how well these parameters could be identified in the simulated data.

## 2.3 Statistical analyses

We used Hotelling's $T^2$ tests for multivariate comparisons (e.g. eigenvectors) between patients and control group and Mann-Whitney U tests to compare non-normal univariate variables. To investigate relations between symptom scores and dynamical features, Spearman's rank correlations were performed. Full statistical tables are in S1 Text, Sec 3 Additional Figures & Tables. To perform group-level multivariate statistics on eigenvectors/singular vectors, we had to align the direction of individuals' vectors by calculating the inner product between an individual's vector and a reference vector (positive vs negative items [1, 1, −1, −1]) and switched sign in case the inner product was <0. For univariate statistics on single dimensions of the vectors we used absolute values of the variable.

# 3 Results

## 3.1 Parameter identifiability

Parameter identifiability in simulated datasets with known ground truth was profoundly affected by a number of experimental aspects. Recoverability improved with time-series length (Fig 2A1–2A3), while it was reduced by increasing measurement noise (Fig 2B1–2B3) and number of mood items acquired (the dimensionality of the observed data; Fig 2C1–2C3). The precision of the inference is determined by the ratio of the sampling rate compared to the decay constant of the measured process. Precise inference is not possible if the underlying process evolves rapidly compared to the sampling frequency (Fig 2D1–2D3).

The recoverability of parameters in dataset 1 without inputs and in dataset 2 with external driving inputs **u** weighted by an input matrix **C** is shown in Fig 3. Specific parameter recovery is relatively poor (Fig 3), though the dominant dynamical features are recovered well (Fig 3C and 3H). Adding estimation of input weights **C** affected inference advsersely. This is because it added a large number of parameters to the estimation, and these are difficult to estimate due to the high dimensionality and extreme sparsity of the input matrix. However, it is important to remember that the apparently better performance on the data without inputs is false to the extent that it does not capture the misestimation due to the missing inputs.

**3.1.1 Capturing empirical features of ESM data.** We next examined whether the Kalman filter could capture empirically observed features in ESM mood data that have a known relationship to depression: summary statistics (mean and (co)variance), inertia (temporal autocorrelation), and instability (root mean square of successive differences).

Each feature was computed for the empirical data in dataset 1 and again for the surrogate data simulated from the Kalman filter fitted to the dataset 1. Features computed on the surrogate data correlated well with features computed from the empirical data (Fig 4; $R(327) \in$ [0.52, 0.99], $p \leq 0.001$). Extracting these statistics from simulated data using a VAR model, we observed a worse recovery in autocorrelation.

**3.1.2 Dynamical differences between patients and controls.** We next examined whether the dynamical parameters were related to depression. In dataset 1, we compared the dynamics matrices **A** between patients ($N = 117$) and controls ($N = 210$). In principle, the dynamics matrix captures how the system moves around the state-space, i.e. how an individual's mood evolves around the fixed point. The diagonal elements of the dynamics matrix define how mood dimensions 'act upon' themselves between two consecutive time points, i.e. how aspects of mood are maintained at a given level over time. Off-diagonal elements describe how a mood item at time $t$ influences another mood item at time $t + \Delta t$, and as such directly captures

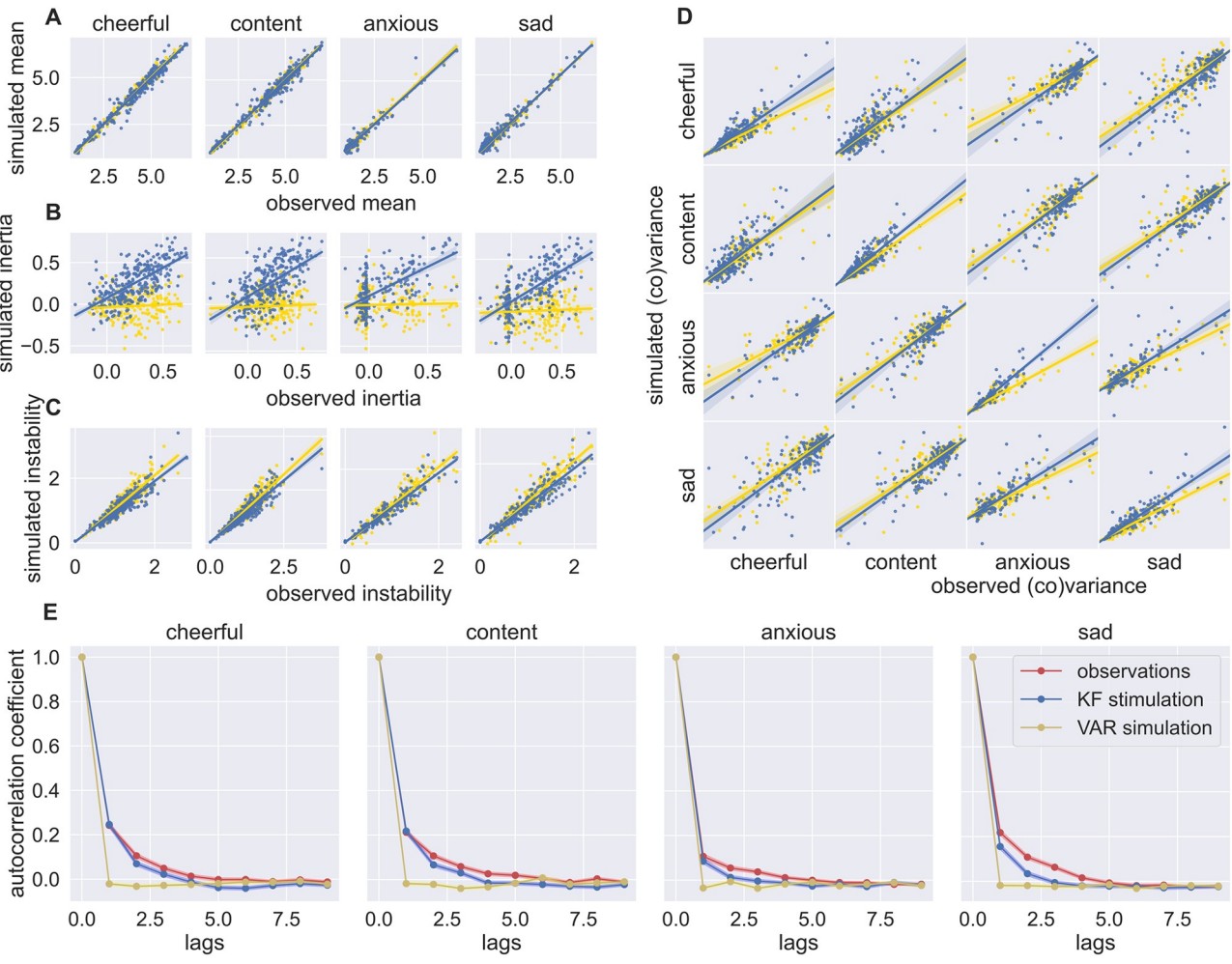

**Fig 4. The Kalman filter captures key qualitative features.** Correlations between dynamical features of empirical individual emotion time-series and simulated time-series using the individual parameter estimates from the state space model (blue) and based on the vector autoregressive (VAR) estimates (yellow). **A)** Empirical mean averaged over time plotted against the mean of the simulated time-series. **B)** Empirical autocorrelation vs the autocorrelation of the simulated emotion time-series. **C)** Empirical root mean squared successive difference capturing frequent and abrupt change vs the root mean squared successive difference of simulated time-series. **D)** Covariance elements of empirical vs simulated data. **E)** Autocorrelation evolving over lags for empirical and simulated time-series.

how different mood items influence each other. As previously noted, in our implementation of the Kalman filter, the latent dynamics evolve in 1-min time steps and the estimated dynamics matrices hence operate at this timescale, too. To provide more meaningful and intuitive insights, we concentrated on hourly dynamics matrices, which revealed the variations occurring within the system within a one-hour time-frame (Fig 5A and 5B). A visualization of how the dynamics matrix elements were modulated by time is in S1 Text Fig A.

Sadness and anxiety decayed less over time and were maintained longer in patients than in healthy controls (Fig 5A and 5B; Diagonal elements of **A** corresponding to anxiety and sadness ratings were higher in patients: anxious: $U = 14367$, $p = 0.011$; sad: $U = 16797$, $p < 0.001$; full statistics in S1 Text Table A). In patients, sadness ratings also had a larger negative impact on positive mood at the next time point (on cheerfulness ratings: $U = 10293$, $p = 0.015$, on content ratings: $U = 9536$, $p < 0.001$). In contrast, ratings of contentedness decayed less ($U = 10276$, $p = 0.014$), enhanced cheerfulness ratings more ($U = 9659$, $p = 0.001$) and decreased sadness

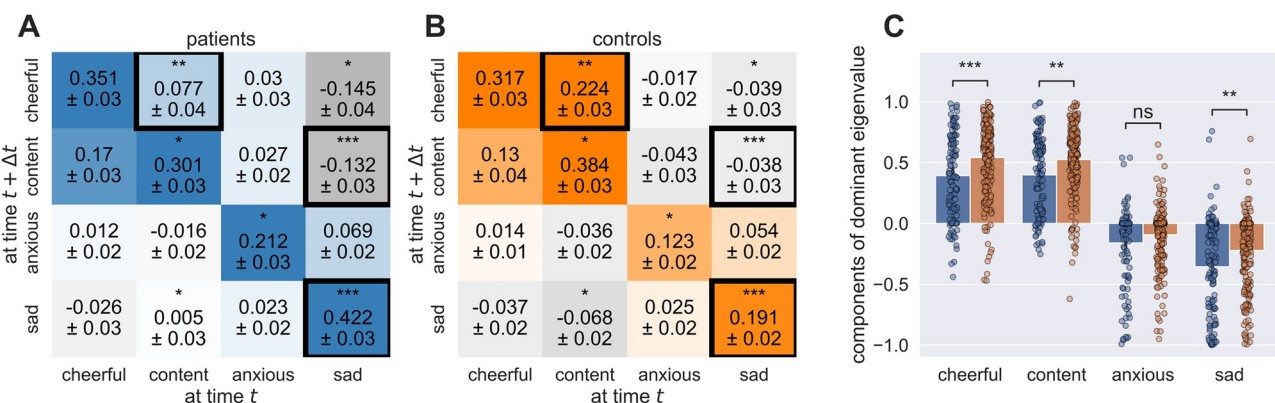

**Fig 5. Differences between patients and controls. A)** shows the estimates of the dynamics matrix elements averaged over patients (M ± SEM). **B)** shows the estimates of the dynamics matrix elements averaged over healthy controls (M ± SEM). The black frame indicates the significant difference between estimates after correcting for multiple comparison. Both averaged matrices reveal a plausible pattern wherein positive items (cheerfulness and contentedness) increase each other while decreasing negative items (anxiety and sadness), and vice versa. **C)** shows the eigenvector corresponding to the dominant eigenvalue for the patient (blue) and control (orange) group. The bars indicate the mean and the dots individual subjects. Significance *≤0.5, **≤0.1, ***≤0.001, ****≤0.0001. Black frames survive correction for multiple comparisons.

ratings more ($U = 14065$, $p = 0.03$) in controls. The differences in the autoregressive parameter of sadness, the influence of contentedness on cheerfulness and of sadness on contentedness ratings survived correction for multiple comparisons (Fig 5A and 5B, black boxes; $p \leq \frac{0.05}{16} \leq 0.003$).

In the VAR model, patients displayed statistically significant elevation in anxiety ratings, whereas the higher sadness rating in patients did not survive multiple comparison correction (see S1 Text Fig B and Table B).

Furthermore, the slowest direction (the eigenvector corresponding to the dominant eigenvalue) differed significantly between patients and controls (Fig 5C and Table 1). We observed no difference in the dominant eigenvalue and thus the speed with which the first eigendirection decays ($U = 13886$, $p = 0.051$) nor the overall stability ($U = 13524$, $p = 0.131$). In both groups, the slowest direction appeared to be a direction where positive and negative mood items diverge. In patients negative valence (averaged over the items anxious and sad) was significantly higher than in controls ($U = 9379$, $p < 0.001$), whereas positive valence (averaged over the items cheerful and content) was significantly higher in controls compared to patients ($U = 9048$, $p < 0.001$). However, the absolute difference between valence, i.e. how much positive and negative mood items diverge, was not different ($U = 11096$, $p = 0.147$).

**Table 1. Comparing the components of the dominant eigenvalue (slowest eigenvector direction) of the dynamics matrix between patients and healthy controls using a Hotelling T$^2$ test.** The eigendirection corresponding to the dominant eigenvalue significantly differed between patients and controls. Single post-hoc test were performed to compare the slowest eigendirection of single mood items. We report mean (M) and standard error of the mean (SEM) ofthe components of the dominant eigenvalue and the results of Mann-Whitney U tests comparing them between patients and healthy controls.

| | | T$^2$ | Fstats | pvalue |
|---|---|---|---|---|
| components of dominant eigenvalue | | 36 | 8.9 | $< 0.001$ |
| | patients (M ± SEM) | controls (M ± SEM) | U | pvalue |
| cheerful | 0.39 ± 0.17 | 0.54 ± 0.15 | 9215 | <0.001 |
| content | 0.4 ± 0.16 | 0.53 ± 0.12 | 9599 | 0.001 |
| anxious | -0.16 ± 0.16 | -0.09 ± 0.12 | 11654 | 0.442 |
| sad | -0.36 ± 0.2 | -0.22 ± 0.15 | 9738 | 0.002 |

Overall, the dynamical interactions in controls was shifted more towards a positive direction and the dynamical interactions of patients moved equally into the negative direction (Fig 5C). Refer to Table C in S1 Text for a complete overview of the differences in dynamical features between patients and healthy controls.

**3.1.3 Linking individual differences in dynamics and depressive symptoms.** We next attempted to replicate the association with depression symptoms qualitatively as correlations between dynamical features and depressive symptoms in dataset 2. A measure of depression was derived from the SCL-90 [74]. Importantly, this dataset also contains measures of participants' environmental context: what they were doing, where they were and whom there were with. We used this information as external inputs to the dynamical system.

Baseline depression scores were positively correlated with the diagonal elements of the dynamics matrix for negative mood items (anxious$_t$ → anxious$_{t+1}$: $r_{\mathrm{spear}} = 0.25$, $p \leq 0.001$, sad$_t$ → sad$_{t+1}$: $r_{\mathrm{spear}} = 0.26$, $p \leq 0.001$; multiple comparison $p \leq \frac{0.05}{16} \leq 0.003$; cf Table 2). This qualitatively replicates the group effects: depression is associated with stronger maintenance of negative mood items, even when controlling for the effect of inputs.

In a VAR model that does not account for unequal time intervals, the association between persistence of sadness and depression score was not seen ($r_{\mathrm{spear}} = 0.06$, $p = 0.265$; Table 2).

The most intrinsically stable combination of mood items was associated with higher depression score, with the stability of items anxious ($r_{\mathrm{spear}} = 0.19$, $p \leq 0.001$) and sad ($r_{\mathrm{spear}} = 0.13$, $p = 0.005$) increasing with depression scores and that of cheerfulness decreasing ($r_{\mathrm{spear}} = -0.14$, $p = 0.005$). Depression symptoms correlated positively with both the dominant eigenvalue ($r_{\mathrm{spear}} = 0.13$, $p = 0.006$) and the overall stability (the determinant of the dynamics matrix; $r_{\mathrm{spear}} = 0.30$, $p \leq 0.001$) suggesting a more stable mood system in depressed states overall. Refer to Table D in S1 Text for a complete overview of the links between dynamical features and depression score.

Depression symptoms also related to controllability. Inferring input weights (**C**) enabled us to investigate the influence of external inputs on each emotion item and to examine how controllable moods are. The most controllable direction of the controllability Gramian was associated with symptoms of depression, pointing less in a positive valence (cheerfulness: $r_{\mathrm{spear}} = -0.13$, $p = 0.008$, content: $r_{\mathrm{spear}} = -0.15$, $p = 0.002$) and more in a negative valence (anxiety: $r_{\mathrm{spear}} = 0.3$, $p \leq 0.001$, sadness: $r_{\mathrm{spear}} = 0.13$, $p = 0.006$). The controllability of the direction itself was not related to symptoms ($r_{\mathrm{spear}} = -0.03$, $p = 0.487$). External factors are more effective in eliciting negative mood and less in eliciting positive mood in participants with higher depression scores.

Inference of the input weight matrix **C** in principle allows examination of the impact of external inputs on moods, controlling for the dynamics of the mood items themselves and for their interactions. As the high-dimensional input weight estimates are noisy, any findings should be interpreted with caution. Nevertheless, being at home (input weight ($M \pm SEM$): $0.31 \pm 0.08$, $T = 3.76$, $p \leq 0.0005$; corrected for multiple comparison p $\leq \frac{0.05}{\text{\#of input weights}} \leq 0.0005$), being in other places outdoors (input weight ($M \pm SEM$): $0.35 \pm 0.09$, $T = 3.67$, $p \leq 0.0005$), and being at work (input weight ($M \pm SEM$): $0.35 \pm 0.08$, $T = 3.52$, $p \leq 0.0005$), increased sadness ratings at the group level. The weighting on anxiety ratings increased with depressive symptoms ($r_{\mathrm{spear}} = 0.24$, $p \leq 0.001$), suggesting that in more vulnerable individuals, the environmental context had a larger absolute effect on anxious symptoms.

**3.1.4 Dynamical changes within an individual.** Finally, we turned to the question of whether the between-individual effects might also be visible within an individual as they move from well to unwell states. This was examined in the third dataset with extended data from an

**Table 2. Comparing links between symptoms and estimates from the Kalman filter with those from the vector autoregressive (VAR) model.** We found a significant association between the decay of sad over time derived from our approach and depressive symptoms, both in the dynamics matrix in minutes and hours. This effect was not present when looking at the autoregressive weight of sad derived from a standard VAR model.

| | dynamics matrix (min) | | dynamics matrix (h) | | VAR | |
|---|---|---|---|---|---|---|
| | $r_{spear}$ | p-value | $r_{spear}$ | p-value | $r_{spear}$ | p-value |
| cheerful → cheerful ($a_{11}$) | 0.04 | 0.35 | 0.04 | 0.37 | 0.01 | 0.77 |
| content → cheerful ($a_{12}$) | -0.04 | 0.43 | -0.04 | 0.42 | -0.06 | 0.25 |
| anxious → cheerful ($a_{13}$) | -0.09 | 0.05 | -0.09 | 0.05 | -0.09 | 0.09 |
| sad → cheerful ($a_{14}$) | -0.04 | 0.45 | 0.03 | 0.48 | 0.04 | 0.44 |
| cheerful → content ($a_{21}$) | 0.06 | 0.19 | 0.05 | 0.30 | 0.02 | 0.66 |
| content → content ($a_{22}$) | -0.02 | 0.70 | -0.01 | 0.83 | -0.04 | 0.41 |
| anxious → content ($a_{23}$) | -0.10 | 0.05 | -0.10 | 0.03 | -0.10 | 0.05 |
| sad → content ($a_{24}$) | -0.06 | 0.20 | 0.01 | 0.88 | -0.03 | 0.62 |
| cheerful → anxious ($a_{31}$) | -0.02 | 0.67 | -0.02 | 0.62 | -0.04 | 0.40 |
| content → anxious ($a_{32}$) | -0.09 | 0.05 | -0.02 | 0.71 | -0.05 | 0.32 |
| anxious → anxious ($a_{33}$) | 0.25 | <0.001 | 0.25 | <0.001 | 0.18 | <0.001 |
| sad → anxious ($a_{34}$) | -0.05 | 0.28 | 0.09 | 0.05 | -0.03 | 0.52 |
| cheerful → sad ($a_{41}$) | 0.02 | 0.69 | 0.00 | 0.96 | 0.00 | 0.98 |
| content → sad ($a_{42}$) | -0.16 | 0.00 | -0.07 | 0.17 | -0.00 | 0.93 |
| anxious → sad ($a_{43}$) | -0.14 | 0.00 | 0.04 | 0.40 | 0.10 | 0.06 |
| sad → sad ($a_{44}$) | 0.28 | <0.001 | 0.26 | <0.001 | 0.06 | 0.23 |

individual before and after discontinuation of antidepressant medication and transition from a less to a more severely depressed state. One separate dynamical systems was estimated for each of three periods: 1) the 4 week period prior to antidepressant discontinuation, 2) the 10 week period during the discontinuation, and 3) the 10 week period after transition to a depressed state. We ensured interpretability of the resulting **C** matrix as it depicted the mood changes corresponding to specific inputs (see S1 Text Fig C).

The most stable eigenvector directions remained similar over the course of the three phases (Fig 6A). A previous study found evidence for an increase in autocorrelation before the transition (averaged over all mood items [75]) suggestive of critical slowing down before transitioning. Our findings mirrored this qualitatively, showing an increase in overall stability, as measured by the determinant of the dynamics matrix, between the baseline and discontinuation phases. This increase persisted after the transition (baseline: 0.84, discontinuation: 0.88, post-transition: 0.88).

The most controllable directions varied across different phases (Fig 6B). Singular vectors, similar to eigenvectors, can have both positive and negative components, indicating the direction and magnitude of the effect in each coordinate, with the ability to switch signs. At baseline, all mood items were aligned in the same direction, making it more difficult to control negative and positive items separately. During the discontinuation phase, the most controllable direction shifted towards more positive items, while negative items became almost uncontrollable. After transitioning to a worse state, the most controllable direction pointed towards a more negative direction, particularly with respect to the "sad" item, and less in a positive direction. Additionally, singular values indicate how easily the system states can be controlled by inputs, with larger values implying easier control and less input energy required to influence the system's dynamics along those directions. We observed that how controllable the most controllable direction was appeared to decrease over the three phases (baseline: 4.2, during

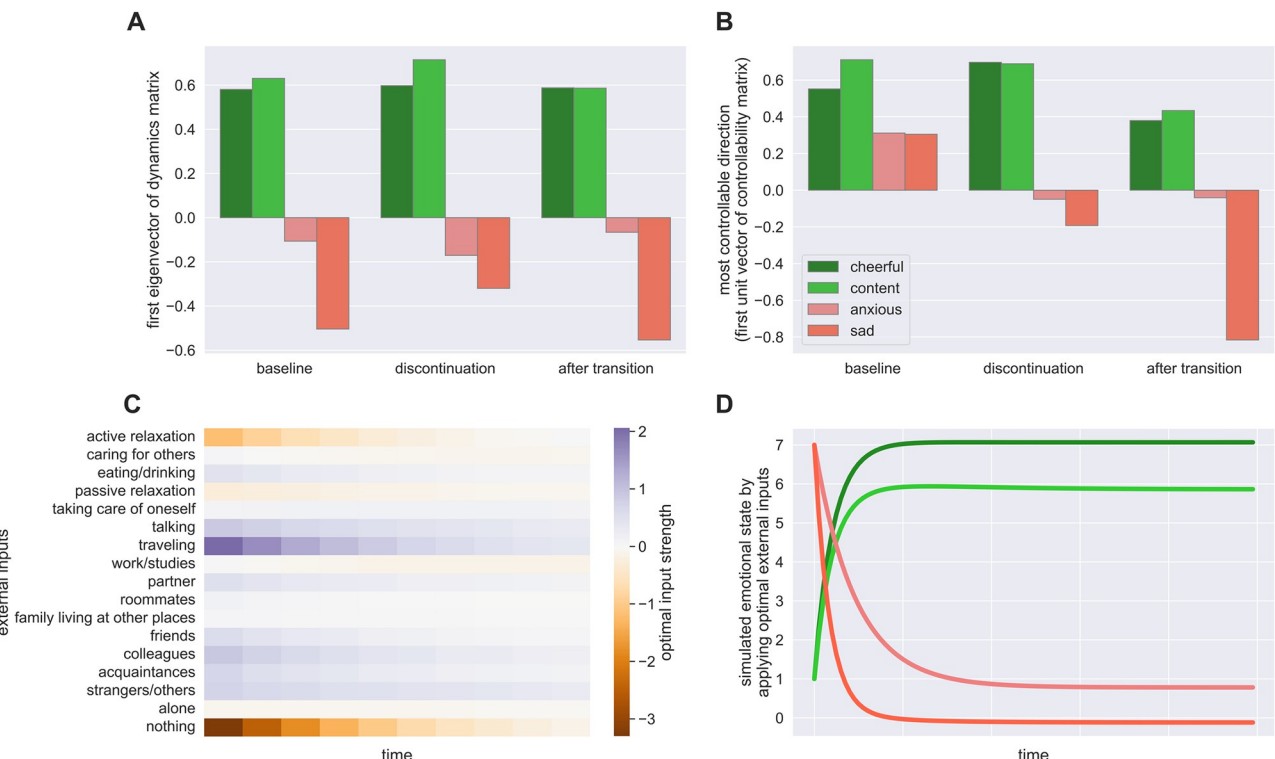

**Fig 6. Transitioning between emotion states. A)** shows the eigenvectors corresponding to the dominant eigenvalue of the dynamics matrix for the three different phases. **B)** shows the left singular vectors corresponding to the dominant singular value of the controllability matrix. **C)** shows the needed input strength to steer the system to a desired state where negative emotions are low and positive emotions are high. **D)** shows the simulated mood states based on the optimal input.

discontinuation: 1.48, after transition: 1.07), which was expected since more stable systems are less controllable.

Might it be possible to identify a sequence of external inputs that would best direct this particular individual's emotional state to a desired target state? We worked with the estimated dynamics after the transition, and used optimal control theory to infer the input sequence $\mathbf{u}_t$ required to steer the system to a "healthy" state with high positive and low negative ratings. The controllability of the system indicated that the trajectories of different valences should be moved apart easily (Fig 6B). Indeed, the optimal input rapidly elicited the target state (Fig 6C and 6D). The input factors "doing nothing" and "traveling" demonstrated the most significant impact but had divergent effect. In this example, a suitable strategy for improving this person's state might have involved increasing travelling while reducing periods of inactivity.

## 4 Discussion

New smartphone technologies allow high-frequency longitudinal measurements of mood and symptom data, enabling the investigation of changes over time in addition to stable momentary states [28, 32]. Currently, vector autoregressive models are the standard method to analyze individuals' variability in experience sampling time series [43]. The approach presented here addresses several limitation of vector autoregressive models by considering the issue of unequal time intervals; allowing for noise in the observations and missing data; allowing for inputs; and providing access to a rich toolbox of methods to estimate and optimize control.

We showed that rich dynamical features of relevance to mental health can be inferred with some accuracy in typical experience sampling scenarios, though aspects such as data dimensionality, the number of samples, the noise regime, and, most significantly, the temporal sparsity have substantial effects.

In three empirical datasets, we found that dynamical mood features 1) distinguished patients with depression from healthy controls and 2) were related to individual variability in depressive symptoms. Our results suggest that the longer-term dynamics of mood systems tend to promote positive mood more in healthy controls and negative mood more in patients with depression. This pattern was also more prominent in individuals reporting feeling worse in a general population sample. Moreover, the speed at which the slowest direction evolved was related to symptoms, i.e. the worse a person feels, the more slowly their mood decays over time, likely driven by an increase in negative mood components. The link between the stability of individuals' mood systems and self-reported depressive scores is consistent with previous research showing an increase in inertia in negative mood items in individuals with depression [42, 76–79].

External inputs are another key factor when investigating dynamic systems and transitions within them. Essentially, neglecting the current context in which emotions evolve may result in inaccurate conclusions about the underlying affective system. In fact, recent studies have established that external factors significantly influence the fluctuations of emotional states in lab experiments [50–53, 80]. For instance, inputs can mask or alter the apparent dynamics of the system. Vanhasbroeck and colleagues [50] have shown that nonlinearity observed in affective time series in some individuals was the result of external inputs rather than an underlying nonlinearity in affect. To ensure accurate understanding of the affective system, it is crucial to consider the relationship between affect dynamics and the immediate environment.

In two of the datasets analysed here, information about participants' context at the time of the measurement was provided, which allowed us to expand our model to a control system. The Kalman filter performance worsened when including this contextual information as external inputs, which may seem counterintuitive as more data (inputs) should improve the model's accuracy. However, the inclusion of data also requires extending the model and adding a large number of parameters. The ratio of additional parameters to additional data is not, unfortunately, conducive to more precise inference. We note, though that the apparently better performance without inputs is misleading because it incorrectly ascribes change driven by external inputs to internal dynamics. This highlights the importance and challenge of effectively measuring emotions and the immediate context.

We examined the inferred system's controllability and found that depression specifically was related to the most controllable emotion combination, showing the same pattern as the slower dynamics. More interestingly, we showed on a group level which emotions were mainly targeted by which environmental influences and that the overall impact on anxiety was related to depression. We posit that a control-theoretic approach holds promise as a foundational framework for the creation of personalized interventions. However, it is important to note that it seems infeasible to account for the entire environmental context based on sampling momentary experiences. Additionally, context-dependent effects likely vary in terms of effect delay, for instance sport might have an immediate influence whereas other activities may have more delayed effects.

Moreover, we demonstrated how the system could be controlled, using inputs to steer emotions to the desired state effectively. This approach allowed us to simulate different input trajectories and may help to identify therapeutical interventions tailored to individuals' affective systems. Overall, studying symptoms directly in a temporally dependent context may increase the relevance for clinical applications. For instance, inferring which symptoms and/or inputs

are most effective and least costly to intervene on could substantially impact personalized therapy approaches for mental illnesses. Other studies [81–84] outlined the use of optimal control for individualized interventions in psychology emphasising the promising potential of control theory in predicting when prevention and intervention have the highest probability of success.

To conclude, how do our findings relate to theories of psychopathology? Depression is characterized by a predominance of negatively valenced, low-arousal emotions such as sadness and lethargy, with a well-documented persistence of these negative mood states (high emotional inertia) [42, 76]. In our study, we found that individuals with depression not only experience more stable patterns of negative mood but also exhibit less emotional flexibility. This observation sits well with cognitive theories of depression, which propose that ruminative thought patterns contribute to this affective inertia, making it difficult for individuals to transition out of negative emotions [77]. Negative affect can persist even in the presence of potentially mood-enhancing events. This is understandable because we observed that in individuals with higher levels of depression the direction requiring the least input effort tends to shift more towards negative moods and less towards positive moods. Therefore, it would require significant effort to achieve a positive mood shift. Behaviorally, reduced engagement in rewarding activities—a core aspect of behavioral activation theory—further exacerbates this issue, as individuals with depression are less likely to seek out or participate in activities that could enhance their mood [42].

## 4.1 Limitations

Using the Kalman filter in our study presents several limitations that warrant consideration for future research. Firstly, while the latent dynamical trajectories are often the main interest [85], we were interested in the connection weights between different mood items and therefore enforced an interpretable setting with a diagonal observation matrix. This reduced the power for estimating complex systems. However, the computational complexity increases quadradically in the state space size, and as the parameter optimization is non-convex, only local optima can typically be found. Even in the reduced model with a diagonal observation matrix, we still observe over/underestimation of dynamic parameters and noise variances due to well-described tradeoffs in the literature. The tradeoff originates from the sum in the marginal likelihood where the dynamics matrix and both noise terms are nested. In order to improve recoverability in higher-dimensional systems, parameters need to be constrained using e.g. lasso-restrictions for the input weights. Another potential approach consists in using multi-level (hierarchical) models. However, [11] also reported overestimated error variances and lower dynamic parameters using a hierarchical state-space model. Furthermore, the assumption of Gaussian error distribution is incorrect as the mood ratings are bounded by being acquired on a Likert scale. We also used binarized control inputs, which can also affect system behavior and may not be optimal.

Next, we assumed that parameters are time-invariant. Estimates of the dynamic characteristics for each individual relied on the entire time-series. This implies that the parameters are constant across all time points. Yet, we have recently found that parameters do change over time, e.g. due to psychotherpeutic inputs [86]. Changes could be incorporated via a higher level of discrete states with distinct dynamics, which would increase the model complexity but also enable the investigation of changes in the dynamics over time. One specific extension would be the examination of switching Kalman filters.

Due to the models' high flexibility, a large amount of data is needed to obtain reliable estimates, particularly with high-dimensionsal observations. However, the number of observations in ESM data is limited, and the data is noisy. Completing surveys multiple times per day

for a more extended period imposes a burden on participants constraining the sample sizes. It might be that our model is too complex for the data at hand. As such, finding the suitable complexity of a model is essential when trying to model affect dynamics. However, it is likely that the processes underlying mood are extremely complex in nature. It could be useful to use a model selection approach to determine the optimal level of complexity for the model. This could be done by comparing the performance of different models with varying levels of complexity, and selecting the one that best fits the data while also being interpretable.

Recent work has shown that simple features of experience sampling data such as the mean and variance of affective emotions are the main predictors of psychological well-being, and complex dynamics and network features may not add additional information [38]. To investigate whether the dynamical features examined here add information above mean and variance was not the focus of this paper but should be addressed in further research. Notably, we know that dynamical features from a linear system without external inputs are strongly linked to the mean, as the fixed point of the model converges to the mean if noise is small. Therefore, our approach is unlikely to increase predictive power. However, here we were mainly interested in a linear model giving us an insight into the parameters characterizing the dynamical system, and we were aware that a linear Kalman filter is not powerful enough to reproduce complex dynamical phenomena. Hence, future research should consider including non-linearity in the model to capture complex dynamical phenomena, like multiple fixed points, limit cycles, or chaos [9].

Finally, we emphasize that while the Kalman filter is a generative model, it remains a descriptive model. It is limited, as noted above, in its expressivity, and its simplistic parameters hide complex underlying cognitive and neurobiological processes which will require richer data and models to characterise and understand.

## 4.2 Conclusion

The creation of models for clinical applications demands the incorporation of individualized frameworks. However, the development and validation of such mathematical models often require extensive datasets encompassing multiple subjects. Moreover, individual time series frequently span various phases of the illness, making stability assumptions impractical. This presents a paradox and a significant constraint in the longitudinal utilization and modeling of experience sampling data.

## Supporting information

**S1 Text. S1 Supplementary Material.**
(PDF)

## Acknowledgments

We would like to express our gratitude to Claudia Menne-Lothmann, Jeroen Decoster, Dina Collip, Marieke Wichers, and Jim van Os for their participation in the TwinssCan data acquisition.

## Author Contributions

**Conceptualization:** Jolanda Malamud, Quentin J. M. Huys.

**Data curation:** Sinan Guloksuz, Ruud van Winkel, Philippe Delespaul, Marc A. F. De Hert, Catherine Derom, Evert Thiery, Nele Jacobs, Bart P. F. Rutten.

**Formal analysis:** Jolanda Malamud.

**Funding acquisition:** Quentin J. M. Huys.

**Methodology:** Jolanda Malamud.

**Project administration:** Jolanda Malamud.

**Software:** Jolanda Malamud.

**Supervision:** Quentin J. M. Huys.

**Validation:** Jolanda Malamud.

**Visualization:** Jolanda Malamud.

**Writing – original draft:** Jolanda Malamud.

**Writing – review & editing:** Sinan Guloksuz, Ruud van Winkel, Philippe Delespaul, Marc A. F. De Hert, Catherine Derom, Evert Thiery, Nele Jacobs, Bart P. F. Rutten, Quentin J. M. Huys.

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
