## [Decision Letter · Decision Letter 0]

17 Jun 2024

Dear Ms Malamud,

Thank you very much for submitting your manuscript "Characterizing the dynamics, reactivity and controllability of moods in depression with a Kalman filter" for consideration at PLOS Computational Biology.

As with all papers reviewed by the journal, your manuscript was reviewed by members of the editorial board and by several independent reviewers. In light of the reviews (below this email), we would like to invite the resubmission of a significantly-revised version that takes into account the reviewers' comments.

We cannot make any decision about publication until we have seen the revised manuscript and your response to the reviewers' comments. Your revised manuscript is also likely to be sent to reviewers for further evaluation.

Sincerely,

Christoph Mathys

Academic Editor

PLOS Computational Biology

Andrea E. Martin

Section Editor

PLOS Computational Biology

Reviewer's Responses to Questions

**Comments to the Authors:**

Reviewer #1: This paper describes the application of a Kalman filter to model individual differences in self-reported mood dynamics. This is an interesting and clearly written paper showing that a new analytic approach can provide novel insights and incorporate additional information into dynamical systems analyses of mood.

Comments/concerns:

1) The authors briefly cover the use of summary statistics and VAR models to analyze mood dynamics, but does not review any previous work on state space models applied to this sort of data. Have previous studies used Kalman filter or similar state space approaches to study mood dynamics?

2) One motivation the authors provide in the introduction for using model-based analyses is that summary statistics approaches “do not explain the underlying data generation process”. Do the authors claim that their model represents the true underlying process generating emotion dynamics?

3) Why did the authors choose minutes as the time steps (versus hours, 15 minute intervals, etc.), given that this leads to very sparse observations? What effects does this choice have on the data, and is there information about the time course of emotion dynamics that can inform the optimal time step?

4) The parameter recovery analyses are useful to understand the limitations of the analyses. With more observations, does recovery improve (similar to the first column in figure 2)? Do these analyses provide any recommendations when designing studies to ensure good recovery?

5) The controllability analyses are interesting but I had a hard time following the interpretation. How can there be negative controllability (in figure 6B)? I am also having trouble connecting the data in figure 6B to the description in the paragraph below (“how controllable the most controllable direction was appeared to decrease over the three phases (baseline: 4.2, during discontinuation: 1.48, after transition: 1.07))”.

Reviewer #2: Thank you for the opportunity to review this manuscript by Malamud and colleagues, which applies a venerable state-space modelling framework (the Kalman filter) to the question of affect dynamics in depression. Three extant datasets were re-analysed using this method, and the results serve as a very compelling proof-of-concept for the application of this analytic framework to this type of data. I was very impressed by the manuscript in general, and have only several relatively minor comments below requesting further elaboration of several aspects of the results.

1. I agree with the manuscript that an important potential bonus of analysing affect dynamics within the state-space modelling framework is the ability of these models to account for the effects of external inputs on subjective affect. Given that potential advantage, therefore, I found it striking that the manuscript found that adding the estimation of input weights to the KF negatively affected inference overall. This is an important point, and should be discussed in more detail in the results (and potentially in the Discussion). In particular, I think more space should be given to discussing exactly which results point to this conclusion (and how), as well as a discussion of the circumstances under which this conclusion would be expected to hold. For instance, is it invariably the case that estimating input weights will negatively affect inference? What about if the subject is placed in an experimental environment where the inputs are exactly controlled and known (e.g., the in-lab modelling paradigm popularised by Robb Rutledge over the last decade or so). No need to do any analyses or simulations to answer these questions, but on reading through I felt it would be important for others to know whether the adverse effects of modelling input are a general feature of the Kalman filter or something more specific to the datasets that were analysed.

2. Very little of the manuscript's discussion is concerned with interpreting the clinical significance of results for theories of depression. I appreciate that this is not the primary focus of the manuscript, but even so I feel that the authors are well placed here to draw some links between their findings and the extant literature on affect dynamics in depression (e.g., work on inertia of affect in depression). I think it would be helpful for readers to have access to an expanded discussion of these effects in the context of psychiatric theory.

Minor points:

- Typo on page 7, line 171: the manuscript uses the word "punctuate" rather than "punctate", which I think is what is intended

- The labelling of subplots in Figure 1 is rather confusing; for instance, subplot B itself includes two subplots themselves labelled a and b. This has the potential to confuse readers and a more unambiguous labelling system should be chosen.

**Have the authors made all data and (if applicable) computational code underlying the findings in their manuscript fully available?**

Reviewer #1: **No: **datasets not publicly available

Reviewer #2: **No: **The URL for a GitHub repository containing data and analysis code links to a repository that does not seem to exist.

PLOS authors have the option to publish the peer review history of their article (what does this mean?). If published, this will include your full peer review and any attached files.

Reviewer #1: No

Reviewer #2: No
---

## [Decision Letter · Decision Letter 1]

4 Sep 2024

Dear Ms Malamud,

We are pleased to inform you that your manuscript 'Characterizing the dynamics, reactivity and controllability of moods in depression with a Kalman filter' has been provisionally accepted for publication in PLOS Computational Biology.

Best regards,

Christoph Mathys

Academic Editor

PLOS Computational Biology

Andrea E. Martin

Section Editor

PLOS Computational Biology

Reviewer's Responses to Questions

**Comments to the Authors:**

Reviewer #1: I thank the authors for their responses. I have no further comments.

Reviewer #2: The authors have addressed all my concerns in this revision.

**Have the authors made all data and (if applicable) computational code underlying the findings in their manuscript fully available?**

Reviewer #1: None

Reviewer #2: Yes

PLOS authors have the option to publish the peer review history of their article (what does this mean?). If published, this will include your full peer review and any attached files.

Reviewer #1: No

Reviewer #2: No

---

## [Editor Report · Acceptance letter]

15 Sep 2024

PCOMPBIOL-D-24-00523R1 

Characterizing the dynamics, reactivity and controllability of moods in depression with a Kalman filter

Dear Dr Malamud,

I am pleased to inform you that your manuscript has been formally accepted for publication in PLOS Computational Biology. Your manuscript is now with our production department and you will be notified of the publication date in due course.

With kind regards,

Jazmin Toth
